# The Effect of a *Pex3* Mutation on Hearing and Lipid Content of the Inner Ear

**DOI:** 10.3390/cells11203206

**Published:** 2022-10-13

**Authors:** Rafael M. Kochaj, Elisa Martelletti, Neil J. Ingham, Annalisa Buniello, Bebiana C. Sousa, Michael J. O. Wakelam, Andrea F. Lopez-Clavijo, Karen P. Steel

**Affiliations:** 1Wolfson Centre for Age-Related Diseases, King’s College London, Guy’s Campus, London SE1 1UL, UK; 2Lipidomics Facility, The BBSRC Babraham Institute, Cambridge CB22 3AT, UK

**Keywords:** peroxisome disorders, hearing loss, lipidomics, mouse mutant, synaptic defect

## Abstract

Peroxisome biogenesis disorders (due to *PEX* gene mutations) are associated with symptoms that range in severity and can lead to early childhood death, but a common feature is hearing impairment. In this study, mice carrying *Pex3* mutations were found to show normal auditory development followed by an early-onset progressive increase in auditory response thresholds. The only structural defect detected in the cochlea at four weeks old was the disruption of synapses below inner hair cells. A conditional approach was used to establish that *Pex3* expression is required locally within the cochlea for normal hearing, rather than hearing loss being due to systemic effects. A lipidomics analysis of the inner ear revealed a local reduction in plasmalogens in the *Pex3* mouse mutants, comparable to the systemic plasmalogen reduction reported in human peroxisome biogenesis disorders. Thus, mice with *Pex3* mutations may be a useful tool to understand the physiological basis of peroxisome biogenesis disorders.

## 1. Introduction

Peroxisomes are intracellular, membrane-bound organelles that play a key role in lipid metabolism and the control of reactive oxygen species [1]. Their numbers can be adjusted as a response to local stresses, thus playing a role in maintaining cellular redox homeostasis. Peroxisomes interact closely with the mitochondria [2,3,4] and can be formed by self-division or by budding off from the endoplasmic reticulum [5,6]. A series of peroxins (encoded by *PEX* genes), including Pex3, import peroxisomal proteins into the new peroxisome. Pex3 can be found in the membrane of both the endoplasmic reticulum and peroxisomes, and acts as a docking site for Pex19 in importing peroxisome membrane proteins (PMPs) [7].

Mutations in *PEX* genes lead to a spectrum of diseases called peroxisome biogenesis disorders (PBD), the most severe of which is Zellweger syndrome, which includes hypotonia, renal and liver defects, seizures, developmental delay, and often leads to early childhood death [8]. Depending on the specific mutation, individuals carrying *PEX* mutations can show milder effects, but hearing loss is a frequent feature of people on the Zellweger spectrum [9,10]. *PEX3* mutations can result in severe or moderate Zellweger spectrum disorder. There are no obvious domains of the PEX3 protein that associate with particular phenotypes, but relatively few have so far been reported. These include nonsense (R300X) and missense mutations (G331R, D347Y), a 1 bp insertion, the deletion of an exon leading to a frameshift and the loss of a splice acceptor site. The phenotypes ranged from very severe with death soon after birth to survival into middle age. Unsurprisingly, the missense mutations tended to be the mildest in terms of impact on phenotype [11,12,13,14,15,16,17]. A common genomic variant within the *PEX3* gene was significantly associated with audiometric thresholds at 4 kHz in the unselected 1958 British Birth Cohort (*p* = 0.000186), suggesting that this gene may play a wider role in hearing loss in the human population [18].

Most *Pex* gene mutations in the mouse lead to early death, but *Pex3^tm1a^* homozygotes survive, making them a useful tool to study the role of peroxisomes in disease. The *Pex3^tm1a^* mouse mutation causes hearing impairment at high frequencies, as revealed by raised Auditory Brainstem Response (ABR) thresholds in homozygotes at 14 weeks old [18].

In the current study, the *Pex3^tm1a^* allele was shown to reduce *Pex3* transcription to 16% of normal levels. Homozygous *Pex3^tm1a^* mutants showed progressive, high-frequency hearing loss. At four weeks old, an age when ABR thresholds are significantly raised, the only apparent structural deficit observed was a reduction in the number of intact inner hair cell synapses in the basal turn. Using a tissue-specific conditional approach, the disruption of *Pex3* in the inner ear resulted in increased ABR thresholds across all frequencies, suggesting that Pex3 is required locally within the inner ear for normal function rather than hearing loss resulting from a systemic deficiency. Finally, since the peroxisomes have an important role in lipid metabolism, the lipid profile of *Pex3* mutant mice was explored, revealing a reduction in plasmalogens and other ether-linked lipids in inner ear samples from mutant mice.

## 2. Materials and Methods

**Ethics statement.** Mouse studies were carried out in accordance with UK Home Office regulations and the UK Animals (Scientific Procedures) Act of 1986 (ASPA) under UK Home Office licences, and the study was approved by the King’s College London Ethical Review Committees. Mice were culled using methods approved under these licences to minimize any possibility of suffering.

**Mice.** The *Pex3^tm1a(EUCOMM)Wtsi^* mutant mice (abbreviated to *Pex3^tm1a^*) were generated and maintained on a C57BL/6N genetic background as part of the Wellcome Sanger Institute Mouse Genetics Project [18,19]. In addition to recessively inherited hearing impairment at 14 weeks old, the phenotypic screening of homozygotes reported male infertility, reduced alanine and aspartate aminotransferase levels in males, reduced amylase levels in females, increased platelet volume, increased Treg cell and mature B-cell percentages, and corneal defects. The eye defects included disruption of the basal layer of the corneal epithelium, loose stromal layers with the presence of blood vessels and increased thickness, and fusion of the iris to the lens. [18,20].

The *Pex3^tm1a^* allele contains a large cassette inserted between exons 3 and 4 that interferes with transcription, leading to the knockdown of expression [19,21] (Figure 1A). The inserted cassette contains a β-galactosidase/*LacZ* reporter gene. Further details can be found at www.mousephenotype.org, accessed on 18 August 2022. Both males and females were included in the experiments as no difference in auditory function was noted between them. These mutant mice are available through the European Mouse Mutant Archive (EMMA).

The *Pex3^tm1c^* allele was generated by crossing *Pex3^tm1a^* mutants to *Gt(ROSA)26Sor^tm1(FLP1)/Wtsi^* mice ubiquitously expressing Flp recombinase. The inserted cassette of the *Pex3^tm1a^* allele was excised by Flp-recombinase-mediated recombination between FRT sites, restoring normal transcription of the gene. Exon 4 was retained, flanked by LoxP sites (Figure 1A). Subsequently, the Flp recombinase allele was bred out of the colony.

*Pex3^tm1c^* mice were mated to *Sox10Cre* mice (Tg(Sox10-cre)1Wdr; mixed genetic background) [22] to delete exon 4 by Cre-recombinase-mediated recombination between LoxP sites and to generate a frameshift mutation of *Pex3* in the inner ear and craniofacial neural-crest-derived tissues that normally express *Sox10* [22]. *Sox10* is normally expressed in all cells of the otic placode and vesicle from mid-gestation [23,24,25], so all inner ear cell types should express Cre recombinase in mice carrying *Sox10Cre*. The *Sox10Cre* allele was transmitted only through the maternal line because transmission via the paternal germline causes the systemic activation of Cre recombinase [26].

For most experiments, homozygous mutant mice were used with their wildtype littermates as controls. Heterozygotes showed no difference compared with wildtypes in the auditory phenotypes tested and were sometimes used as controls.

**Genotyping.** Genomic DNA was extracted from pinna skin and used as a template for short-range PCR. *Pex3^tm1a^* mice were genotyped using a common forward primer (5′GCCAAACCATAGCACCAGC3′), a wildtype reverse primer (5′CTTTGTCCTCTTTCTGGGCAC3′) and a mutant-specific reverse primer (5′TCGTGGTATCGTTATGCGCC3′). The resulting band sizes were 399 bp for the *Pex3* wildtype product and 264 bp for the mutant product. The wildtype product was not amplified from the mutant allele because the primers bound to sites that were too far apart for short-range PCR to be successful. The forward primer 5′CAAGATGGATTGCACGCAGGTTCTC3′ and the reverse primer 5′GACGAGATCCTCGCCGTCGGGCATGCGCGCC3′ specific for the neomycin resistance gene in the introduced cassette were also used to detect the presence or absence of the inserted DNA of the mutant allele (Figure 1A) [19,21].

The *Pex3^tm1c^* mice were genotyped using the same primers as for the *Pex3^tm1a^* mice, but in this case the *Pex3* wildtype product could be amplified and was 100 bp larger (499 bp) in comparison with the wildtype allele (399 bp) due to the presence of two loxP sites and one FRT site (Figure 1A). The presence or absence of the *Flp* recombinase gene was determined using a common forward primer 5′AAAGTCGCTCTGAGTTGTTAT3′ with a wildtype reverse primer 5′GGAGCGGGAGAAATGGATATG3′, and with a mutant-specific reverse primer 5′GCGAAGAGTTTGTCCTCAACC3′, resulting in band sizes of 603 bp and 250 bp, respectively.

The conditional *Pex3^tm1d^* allele was genotyped by the presence of both *Pex3^tm1c^* and *Sox10Cre* allele PCR bands. The *Sox10Cre* allele was detected by a 101 bp band PCR using the forward primer 5′GCGGTCTGGCAGTAAAAACTATC3′ and the reverse primer 5′GTGAAACAGCATTGCTGTCACTT3′. The deletion of exon 4 in the *Pex3^tm1d^* allele was confirmed by Sanger sequencing (Source BioScience, Nottingham, UK). Primers from exon 3 (5′AGAGAGGCCTTAATGCAGCA3′) and exon 5 (5′GAACCACCAGCATACACGTG3′) of *Pex3* were used to amplify cDNA from the cochlea.

**Auditory Brainstem Response (ABR) recording.** Mutant mice and their littermate controls at several ages were anaesthetised by the intra-peritoneal injection of 100 mg/kg Ketamine (Ketaset, Fort Dodge Animal Health) and 10 mg/kg Xylazine (Rompun, Bayer Animal Health). The brainstem auditory-evoked potentials were measured as described previously [27,28] in a sound-attenuating chamber fitted with a heating blanket. Subcutaneous recording needle electrodes (NeuroDart; Unimed Electrode Supplies Ltd., Farnham, UK) were inserted on the vertex and overlying the left and right bullae. Responses were recorded to free-field calibrated broadband click stimuli (10 µs duration) and tone pips (5 ms duration, 1 ms onset/offset ramp, fixed phase onset) at frequencies between 3 and 42 kHz, at levels ranging from 0–95 dB SPL in 5 dB steps, at a rate of 42.6 stimuli per second. The evoked responses were digitized, amplified and bandpass-filtered between 300–3000 Hz, and 256 responses were averaged for each frequency/intensity combination to yield a waveform. The ABR thresholds were defined as the lowest stimulus level to evoke a visually detected waveform. The ABR wave 1 amplitude (defined as the peak-to-peak amplitude from the first positive wave deflection (P1) to the first negative wave deflection (N1) of the trace) and ABR wave P1 latency were measured as a function of stimulus level (dB SPL), for 12 kHz, 18 kHz and 24 kHz stimuli, using a custom software application [29].

**Distortion Product Otoacoustic Emissions (DPOAE) recordings.** Measurements of the 2f1-f2 DPOAE component were made in mice aged 4 weeks old, as described previously [30]. Mice were anaesthetised with intra-peritoneal urethane (0.1 mL/10 g body weight of a 20% *w*/*v* solution of urethane in water). The DPOAE probe assembly located within a hollow conical speculum was positioned in the external ear canal of the left ear. Continuous f1 and f2 tones were delivered with frequencies for f2 set to 6, 12, 18, 24 and 30 kHz, with f2 at a frequency 1.2x that of f1 and at 10 dB lower than f1. The f2 stimuli ranged from −10 dB to 65 dB in 5 dB steps. DPOAE responses were detected and processed online using Fast Fourier Transformation to yield a power spectrum containing the f1, f2 and 2f1-f2 DPOAE components. 2f1-f2 DPOAE amplitudes were plotted for each stimulus intensity and the threshold was defined as the lowest stimulus level when the DPOAE amplitude exceeded 2 standard deviations above the recording noise floor.

**Endocochlear Potential (EP) recordings.** Mice were anaesthetised with intra-peritoneal urethane as above and the positive potential within the scala media of the cochlea was measured as described previously [31,32]. A reference electrode (Ag-AgCl pellet) was positioned under the skin of the neck. A small hole was made in the basal turn lateral wall and the tip of a 150 mM KCl-filled glass micropipette was inserted into the scala media. The EP was recorded as the differential potential between the tip of the glass electrode and the reference electrode.

**Quantitative reverse-transcription PCR (qRT-PCR).** RNA was isolated from brains of 4-week-old *Pex3^tm1a^* homozygotes and their littermate controls using TRIzol Reagent^®^ (Sigma Aldrich, cat. No. T9424, St. Louis, MO, USA) according to the manufacturer’s protocol. For the *Pex3^tm1d^* allele, whole cochleae were used instead after collection in RNAlater stabilisation reagent (Thermo Fisher Scientific, cat. No. AM7024, Waltham, MA, USA) and RNA was extracted using the SPLIT RNA extraction kit (Lexogen, cat. No. 008.48, Vienna, Austria). RNA concentration was normalised to that of the lowest sample concentration. Any residual DNA was removed using DNAse I (Sigma-Aldrich, cat.no. AMP-D1), before generating cDNA using Superscript II Reverse Transcriptase (Invitrogen, cat. no. 11904-018, Carlsbad, CA, USA). To quantify the level of *Pex3* expression, cDNA was added to SSOFast Advanced Universal Probes Supermix (BioRad, 1725281, Hercules, CA, USA) and the Pex3 Taqman^®^ probe (Applied Biosystems, Mm_01318747, Foster City, CA, USA) designed to span exons 10 and 11. The *Hprt* gene (a housekeeping gene) was used as an internal reference for both brain and cochlear samples, and for the cochlear samples *Jag1* (expressed in organ of Corti supporting cells) was used as a control for sensory tissues (Applied Biosystems, Foster City, CA, USA; *Hprt* Mm_4351370; *Jag1* Mm_4351372). Each sample was tested in triplicate and the mean fold change between wildtype and homozygous and heterozygous mice was normalised to *Hprt* levels, as a housekeeping gene [33]. The 2^−ΔΔCΤ^ calculation [34] was used to analyse the results. The data were checked for normality and the statistical analyses used were either the one-way ANOVA test or the Mann–Whitney test as appropriate.

**β-Galactosidase staining.** The β-galactosidase reporter gene in the inserted cassette of *Pex3^tm1a^* heterozygotes was used to investigate the distribution of expression, using wildtype littermates as controls. Inner ears of 2-week-old *Pex3^tm1a^* mice were fixed in 4% paraformaldehyde for 1 h with rotation at 4 °C, washed in phosphate-buffered saline and decalcified in 0.1 M ethylenediaminetetraacetic acid (EDTA) overnight. Samples were permeabilised for 30 min with a detergent solution (2 mM MgCl_2_; 0.02% NP-40; 0.01% sodium deoxycholate in PBS, pH7.3). X-gal (Promega, cat.no. V394A) was added 1:50 to pre-warmed staining solution (5 mM K_3_Fe(CN)_6_ and 5 mM K_4_Fe(CN)_6_ in detergent solution), then the inner ears were stained at 37 °C in the dark overnight. Following X-gal staining, the samples were washed, dehydrated and embedded in paraffin wax. Samples were sectioned at 8 µm, counterstained using Nuclear Fast Red (VWR, cat.no. 342094W, Radnor, PA, USA) and mounted using Eukitt quick-hardening mounting medium (Sigma-Aldrich). Sections were imaged using a Zeiss Axioskop microscope connected to AxioCam camera and interfaced with Axiovision 3.0 software.

**Scanning Electron Microscopy.** Cochlear samples from *Pex3^tm1a^* homozygotes and their wildtype littermate controls at 4 weeks old were fixed in 2.5% glutaraldehyde in 0.1 M sodium cacodylate buffer with 3 mM calcium chloride, dissected to expose the organ of Corti, then processed by a standard osmium tetroxide-thiocarbohydrazide (OTOTO) protocol [35]. After dehydration, samples were subjected to critical-point drying, mounted and viewed using a Jeol JSM-7800F Prime field emission scanning electron microscope at 10 kV. The cochlear duct shows a tonotopic organisation, with the basal turn being most sensitive to high frequencies and the apical turn to low frequencies. Therefore, we were careful to examine and compare precise locations along the duct. An overview of the cochlea was imaged to allow calculation of percentage distances along the cochlear duct to superimpose the frequency-place map [36], allowing subsequent imaging of consistent locations across different specimens. Images were assessed by two viewers who were blinded to genotype.

**Synaptic labelling and confocal imaging.** Inner ears from *Pex3^tm1a^* homozygotes and wildtype littermate controls aged 4 weeks were fixed in 4% paraformaldehyde for 2 h and decalcified in EDTA for 2 h. Following fine dissection, the organ of Corti was permeabilised in 5% Tween in PBS for 30 min and incubated in fresh blocking solution (4.5 mL of 0.5% Triton X-100 in PBS and 0.5 mL of normal horse serum) for 2 h at room temperature. The primary antibodies used overnight at room temperature were mouse anti-GluR2 (1:200, MAB397, Emd Millipore, Darmstadt, Germany) and rabbit anti-Ribeye (1:500, 192 103, Synaptic Systems, Göttingen, Germany). After washing in PBS, the samples were incubated for 55 min at room temperature in the dark with the secondary antibodies: goat anti-mouse IgG_2a_ Alexa Fluor488 (1:300, A21131, ThermoFisher Scientific) and goat anti-rabbit IgG Alexa Fluor546 (1:300, A11035, ThermoFisher Scientific). After washing, the samples were mounted using ProLong Gold mounting media with DAPI and stored at 4 °C. Specimens were imaged using a Zeiss Imager 710 confocal microscope interfaced with ZEN 2010 software. A plan-APOCHROMAT 63x Oil DIC objective was used for all the images with 3.0 optical zoom. Brightness and contrast were normalised for the dynamic range in all images. Z-stacks were collected at 0.25 μm spanning all synaptic components and maximum intensity projection images were generated for analysis. The best-frequency areas were determined according to the mouse tonotopic cochlear map described by Müller et al. [36]. The number of ribbon synapses per IHC was quantified by manually counting the co-localised Ribeye and GluR2 puncta in the confocal maximum projection images and dividing them by the number of IHC nuclei (DAPI). An average of six IHCs per image were counted using the cell-counter plugin in Fiji software. The normality of the data was determined using Shapiro–Wilk and equal-variance tests, and then the data were analysed by one-way ANOVA or Mann–Whitney tests as appropriate.

**Lipidomics.** Peroxisomes have an important role in lipid metabolism. Therefore, the lipid profile of *Pex3* mutant mice was explored by liquid chromatography coupled with mass spectrometry (LC–MS) and relative levels of the 671 lipid molecular species examined were calculated. Both left and right whole inner ears of 4-week-old *Pex3^tm1a^* homozygous mutants (2 males and 2 females) and sex-matched littermate wildtype controls were collected by dissection in ice-cold saline, were snap-frozen in liquid nitrogen and stored at −80 °C until lipid extraction. Inner ears were collected at the same time of day to avoid circadian rhythm changes. Samples were extracted using butanol for the detection of sphingosine-1-phosphate and lysolipids, using acidified solvent extraction for detecting phosphatidylinositol phosphates, and using Folch extraction [37] for detecting glycerophospholipids, glycerolipids, free fatty acids, ceramides, cholesterol and cholesterol esters, and sphingomyelins. Thirty-eight lipid subclasses from the glycerophospholipid, sphingolipid, glycerolipid, fatty-acid and sterol categories (Table 1) were analysed using both an Orbitrap Elite (Thermo Fisher Scientific) and a triple quadrupole (SCIEX 6500) mass spectrometer in both positive and negative polarities.

For each lipid subclass, the samples were spiked with one molecular species that was not endogenous to the samples (for example, a molecular species with an odd number of carbons such as PA 35:1). These species were spiked at a known concentration and the area under the curve was obtained from the LC–MS and LC–MS/MS data. The area under the curve for each endogenous species was also obtained and the concentration was calculated in relation to the spiked lipid internal standard, giving the ng of each molecular species for each sample. The non-endogenous lipids used to spike samples were as follows (abbreviations for each subclass are given in Table 1):Phospholipids: PA 35:1; PE 35:1; PS 35:1; PI 37:4; PG 35:1; PC 35:1; CL 56:0.Sphingolipids: SG C17; Cer C17; SM C17.Fatty acids: FA 19:0.Lysolipids: LPA 17:0; LPC 19:0; S1P C17; LPE 17:1; LPG 17:1; LPS 17:1.Neutral lipids: CH d7; CE 17:0; DG 35:1; TG 51:3.

The amount of each lipid was normalised to account for varying sample sizes using the amount of DNA in each sample prior to lipid extraction. The DNA was measured from the aqueous phase generated during lipid extraction. Using a nanodrop, the DNA was measured, and the ng of DNA was determined. Then the ng of each molecular species determined by LC–MS and LC–MS/MS analyses was divided by the ng amounts of DNA, to obtain the lipid amounts as ng/ng of DNA. The totals of each subclass were calculated by the sum of the amounts of each molecular species in that class.

Results were analysed firstly by comparing the mean nanogram quantities in the four mutant samples compared with four wildtypes for all 671 lipid species, and secondly by comparing each mutant sample with its littermate control to obtain a ratio before calculating the mean of the ratios and plotting as Log_2_ plots to compare the fold change for each lipid subclass. Statistical comparison was performed using the paired *t*-test (*p* < 0.005), principal component analysis (PCA), and log_2_ ratio transformation. Further details of the analytical protocols have been published [38,39,40].

## 3. Results

### 3.1. Pex3^tm1a^ Homozygous Mutants Are Viable and Show Reduced Transcription

The *tm1a* allele is designed to interfere with the transcription of the gene by introducing a large cassette into an intron to impair splicing to the downstream exon [21]. In the case of *Pex3^tm1a^*, the cassette inserted in the intron between exons 3 and 4 was large enough to interfere with normal transcription of the gene, leading to reduced transcription. *Pex3* mRNA expression was knocked down to 16% of normal levels in the *Pex3^tm1a^* homozygotes (Figure 1B) so the mutation did not lead to a complete knockout of gene transcription, but the knockdown was enough to lead to the raised ABR thresholds at 14 weeks old, as previously reported [18]. The heterozygotes showed an intermediate level of 58% transcription (Figure 1B). The *Pex3^tm1a^* homozygotes were viable but comprised only 17% of the offspring from the heterozygous intercrosses rather than the expected 25%.

The *LacZ* component of the inserted cassette in the *Pex3^tm1a^* allele allowed us to use a β-galactosidase reporter assay to show cells that would normally express *Pex3*. This revealed strong expression in the spiral ganglion and widely around the cochlear duct, including sensory hair cells and the stria vascularis (Figure 1D–H).

### 3.2. Auditory Responses Develop Normally in Pex3^tm1a^ Mutants but Show Progressive Deterioration

Mice normally start to respond to sounds between 12 and 14 days old, and thresholds improve over the following week. ABRs were recorded at 17 days old and homozygous *Pex3^tm1a^* mutants had only slightly raised ABR thresholds at that age, suggesting a largely normal development of hearing (Figure 2A). All three genotypes continued to develop more mature ABR thresholds between P17 and P21, as is often seen, but the slightly raised thresholds in the mutants at P17 may reflect a delay in this maturation process, which may correspond to the developmental delay reported in children with Zellweger syndrome. However, by three weeks old, the mutants showed an increased threshold at high frequencies compared with wildtype littermate controls, and the raised thresholds persisted at four and eight weeks old (Figure 2B–D). Heterozygotes showed normal thresholds (Figure 2).

The analysis of the ABR waveforms at four and eight weeks old revealed reduced amplitudes and increased latencies of wave 1, which is the earliest neural response from the cochlea, in the mutant mice at 12, 18 and 24 kHz (Figure 2E–J show results at four weeks old).

The ABR represents a neural response to sounds, which could originate in the organ of Corti or the cochlear nerve, while DPOAEs reflect outer hair cell function. To determine if the increase in ABR thresholds was due to a hair cell dysfunction or a neural defect, DPOAEs were recorded to assess the function of the outer hair cells in the cochlea. Thresholds for DPOAEs were raised at high frequencies, reflecting the pattern seen for ABRs, indicating that hair cell function was affected (Figure 2K).

### 3.3. Endocochlear Potential Is Normal

Endocochlear potential is a positive resting voltage in the endolymph of the scala media that bathes the upper surface of the organ of Corti and is actively generated by the stria vascularis on the lateral wall of the cochlear duct. As the stria vascularis expresses *Pex3* (Figure 1D,H), strial function was investigated by recording the endocochlear potential. This showed normal levels in four-week-old homozygous mutants, suggesting that strial function was not affected by the *Pex3* mutation (Figure 2L).

### 3.4. Stereocilia Bundles of Hair Cells Appear Normal in Pex3^tm1a^ Mutants

The inner ear was examined to determine any structural correlates of the raised auditory thresholds. The gross structure of the middle ear and inner ear appeared normal upon dissection, with no obvious malformations in the mutants. We used scanning electron microscopy to analyse the upper surface of the organ of Corti at four weeks old, an age when ABR thresholds were raised at high frequencies, including the 30 kHz best-frequency location. The normal arrangement of three rows of outer hair cells and one row of inner hair cells was observed in both *Pex3^tm1a^* mutants and littermate controls at all locations along the length of the cochlear duct examined (Figure 3A–F). At higher magnifications, stereocilia bundles showed no differences between the mutant and control samples (Figure 3G–J).

### 3.5. Synaptic Defects in Pex3^tm1a^ Mutants

As no abnormalities were found at the upper surface of the organ of Corti, we examined the ribbon synapses between inner hair cells and cochlear neurons (Figure 4), which are the major afferent connections with the brain, in *Pex3^tm1a^* homozygotes and wildtype littermate controls aged four weeks. Pre-synaptic ribbons were labelled with anti-Ribeye antibody and post-synaptic densities with anti-GluR2 antibody. The DAPI labelling of nuclei revealed no obvious gaps in the row of IHCs, supporting the scanning electron microscopy finding of minimal or no IHC loss. The labelled puncta below the IHCs were quantified at five positions corresponding to the best-frequency locations of 12 and 18 kHz, where mutant thresholds were normal, and 24, 30 and 36 kHz, where mutant ABR thresholds were raised. There was no difference in the numbers of ribbons or of post-synaptic densities per inner hair cell at any of the locations. However, in the basal turn of the cochlea, more orphan ribbons were observed, separated from their post-synaptic densities (Figure 4B,C). Pre-synaptic ribbons (labelled by Ribeye in magenta) and post-synaptic densities (labelled by GluR2 in green) were often separated in this region (white arrows in Figure 4C), suggesting that the synapse was no longer connected and functional. The quantification of co-localised pre- and post-synaptic markers indicative of intact synapses revealed a decrease in the number of intact synapses in the basal turn, corresponding to the frequencies most affected by raised ABR thresholds (Figure 4C–E).

### 3.6. Pex3 Is Required Locally for Normal Hearing

To establish if the hearing impairment was due to a local deficiency of Pex3 or to a systemic effect that led to hearing loss as a secondary feature, the conditional allele, *Pex3^tm1c^*, was generated by crossing with a ubiquitous Flp-recombinase-expressing mouse line (Figure 1A). Flp recombinase recombines between FRT sites (green triangles) and deletes the sequence in between, restoring normal transcription by returning the intron to close to its normal size. The *Pex3^tm1c^* homozygotes had normal ABR thresholds at 4, 8 and 14 weeks old (Figure 5A–C). We then introduced a Sox10-driven Cre recombinase transgene to recombine between LoxP sites (red triangles in Figure 1A) and delete exon 4 of the *Pex3* gene in all tissues that would normally express Sox10, which includes all cell types in the inner ear. These mice, in which exon 4 of *Pex3* was deleted in the inner ear (the *Pex3^tm1d^* allele), had severely raised ABR thresholds across all frequencies (Figure 5D–F), indicating that *Pex3* expression is required locally within the inner ear rather than the hearing loss being mediated by a systemic effect. An analysis of the ABR waveforms at four and eight weeks old revealed reduced amplitudes and increased latencies of wave 1 in the *Pex3^tm1c/tm1c^* with *Sox10Cre* mutant mice at 12, 18 and 24 kHz compared with the control mice (Figure 5G–L show results at four weeks old).

In the *Pex3^tm1d^* homozygote inner ears, both pre-synaptic ribbons and post-synaptic densities showed a trend of reduced numbers per IHC, as well as a significantly reduced number of intact synapses at higher frequency locations (Figure 6A–G). This was a more severe effect than that seen in the *Pex3^tm1a^* homozygotes. No sign of missing IHC nuclei was observed in the whole mount preparations examined by confocal microscopy (e.g., Figure 6B,D). Finally, we used qRT-PCR of the inner ear from four-week-old mice to test whether the *Pex3* gene was completely knocked out in the *Pex3^tm1d^* allele (Figure 1C). We found that *Pex3* transcription was reduced to 42% of the wildtype level in the homozygotes, suggesting that mRNA expression was reduced but not eliminated. The sequencing of cDNA from the inner ear indicated that some of the sequence from exon 4 was present in the samples. This suggested that the Cre recombinase activity was not fully efficient in the inner ear tissues tested, leading to mosaicism, with some cells still carrying the *Pex3^tm1c^* allele containing exon 4. To produce a complete knockout, the Cre recombinase needs to be highly efficient as it needs to cut exon 4 from both copies of the *Pex3* gene in every cell. Despite the observed retention of some copies of exon 4 revealed by our sequencing, the overall effect of the Sox10-driven Cre recombinase was enough to lead to raised ABR thresholds across all frequencies.

### 3.7. Lipidomic Analysis Reveals a Reduction in Inner Ear Plasmalogens

A lipidomic analysis was carried out on the inner ear samples. A total of 671 lipid species were examined (Appendix A). The amounts of single molecular species showing a significant (*p* < 0.05, *t*-test) decrease or increase of the mean of the four mutant samples compared with the four wildtypes are shown in Figure 7A–D. Several lipid species were significantly reduced in the *Pex3^tm1a^* homozygotes, including plasmenyl-phosphatidylcholines (P-PC), plasmanyl-phosphatidylcholines (O-PC), plasmenyl-phosphatidylethanolamines (P-PE), plasmanyl-phosphatidylethanolamines (O-PE), and plasmanyl-triacylglycerols (O-TG), while some sphingomyelin species (SM) were increased. O-PE, O-PC, P-PC and P-PE are ether-linked phospholipids, subclasses of the glycerophospholipids (Table 1). P-PC and P-PE are plasmenyl lipids, called plasmalogens.

Using a different approach, each mutant mouse sample was compared with its corresponding sex-matched littermate wildtype control. The resulting ratio for each mouse pair was used to calculate a mean, which was plotted on a log_2_ plot to compare the fold change of the lipid subclasses between the two genotypes (Figure 8A). This analysis revealed a reduction in O-LPE, O-PC, O-PE, O-TG, P-PC and P-PE, again suggesting that reduced levels of plasmalogens were present in the mutant inner ears. In addition, phosphatidylinositol (3,4,5)-trisphosphate (PIP3) was increased in the mutant ears (Figure 8A).

A comparison of the log_2_ fold change for males and females separately showed that the levels of plasmenyl lipids (the plasmalogens P-PE and P-PC), and plasmanyl lipids (O-PE and O-PC) were decreased in both female and male mutants (Figure 8B). Females had increased triacylglycerols (TG) and fatty acids (FA) in comparison to males (Figure 8B).

## 4. Discussion

Mutations of *Pex* genes in the mouse often lead to early lethality. In contrast, *Pex3^tm1a^* homozygotes generally survive, making them a useful tool to study the role of peroxisomes in disease, along with previously reported *Pex7* mutants that can also survive [41,42]. In *Pex3^tm1a^* homozygotes, transcription was found to be knocked down to 16% of normal levels and was associated with progressive high-frequency hearing loss, while heterozygotes with transcription reduced to 58% of wildtype levels had normal auditory thresholds, at least up to eight weeks old. These findings suggest a graded response of auditory function to varying levels of *Pex3* transcription, with 58% of transcript being adequate for normal auditory function while 16% was not enough. The basal turn showed greater sensitivity to Pex3 levels than the apical turn in the homozygous *Pex3^tm1a^* mutants. A more severe impact on ABR thresholds was seen when the normal Pex3 protein was effectively knocked out in the inner ear using a conditional approach.

The combination of normal endocochlear potentials (suggesting normal stria vascularis function) with raised thresholds for ABRs and DPOAE responses implicates the sensory hair cells in the pathology. A middle ear defect was not seen, and the normal thresholds at low frequencies in the *Pex3^tm1a^* mutants do not support a conductive defect, which would be expected to affect all frequencies. At four weeks old, when auditory thresholds were already raised, the organisation of the organ of Corti was normal and stereocilia bundles at the top of hair cells appeared ultrastructurally normal. However, in the basal turn, corresponding to the frequencies showing increased thresholds, synaptic abnormalities were detected, with separation of pre- and post-synaptic puncta more commonly seen in the mutants than in wildtypes. Our finding of *Pex3* expression in spiral ganglion cells using LacZ as a reporter supports an early neuronal defect.

Several studies have highlighted the association between a reduced ABR wave 1 amplitude and a loss of co-localised pre- and post-synaptic markers [43,44]. However, a reduced ABR wave 1 is not necessarily diagnostic of a cochlear synaptopathy, and other pathologies such as inner hair cell or myelination defects can result in the same phenotype [30,45]. A reduced ABR wave 1 amplitude was seen in *Pex3^tm1a^* homozygotes in regions of the cochlea (12 kHz region), where the numbers of inner hair cell synapses were normal and ABR thresholds were normal. It is not surprising that changes in cochlear physiology (amplitudes and latencies of ABR wave 1) can be detected before they become sufficiently advanced to become detectable as an anatomical defect (loss of synapses). The modelling of cochlear responses also suggests that the loss of synapses alone may not lead to increased latencies [46], supporting the suggestion that the loss of synapses in the *Pex3* mutant cochlea may be secondary to synaptic dysfunction.

The effects of the *Pex3^tm1a^* mutation on DPOAE responses may also suggest that a loss of inner hair cell synapses is not the only site of pathology in these mice. The DPOAE thresholds in four-week-old mutants were similar to the ABR threshold elevation patterns, suggesting that both the inner and outer hair cells were dysfunctional. The physiological changes in these mutants may have resulted from a more widespread pathological process in cochlear hair cells rather than only a loss of inner hair cell synapses.

Pex3 has a key role in the formation of mature peroxisomes, and peroxisomes in turn have wide-ranging functions throughout the body, particularly in lipid processing. The impact of *Pex* gene mutations on hearing loss, a consistent feature of human Zellweger spectrum disorders, could be mediated through systemic effects or locally through a requirement of peroxisome activity within the inner ear. Devising a treatment to improve lipid balance in the circulation could be simpler than developing site-specific treatments. To address this question, a conditional approach was used to knockout the *Pex3* gene by deleting a critical exon in inner ear tissues. This led to more severe auditory dysfunction, confirming that Pex3 is required locally in the cochlea for normal hearing. This observation has implications for developing treatments for peroxisome disorders and suggests that systemically improving lipid metabolism may not be effective in reversing hearing loss.

Another protein, pejvakin, has been reported to be involved in hearing impairment via the degradation of peroxisomes [47,48], although another group studying a different pejvakin mutation found that the link with peroxisomes was unclear [49]. Peroxisomes are believed to protect cells from oxidative stress [49,50,51,52,53], and oxidative stress is thought to underlie noise-induced cochlear damage [54] and synaptic damage [55]. A role of peroxisomes in oxidative stress may mediate the synaptic defects and hearing impairment that we report in *Pex3* mouse mutants, and the sensitivity to noise-induced damage is a hypothesis worth following up on as it may lead to further mechanistic insight.

Lipidomics data were successfully collected without pooling samples despite the amount of tissue being limited to two inner ears from each mouse. There was variability in the data, which was likely due to the very small amount of material from each mouse. In both analysis approaches, after comparing four mutants with four controls and looking at ratios between individual littermate pairs, the lipids of the plasmalogens group were clearly decreased in inner ears from the *Pex3^tm1a^* mutants. This pattern was observed in both males and females. These results match the reports of reduced plasmalogen levels in peroxisomal disorders in humans [56,57,58], suggesting that the *Pex3^tm1a^* mutant mouse may be a useful system to examine peroxisomal defects more generally.

We were able to identify long-chain fatty acids up to C24:0, but not longer than this, in the very small samples we obtained from the two inner ears of each mouse. Raised levels of very-long-chain fatty acids (VLCFA; C26:0 and C26:0/C22:0 ratio) in the plasma have traditionally been used as diagnostic for patients with peroxisomal disorders [9,59,60], but recently it has been suggested that C26:0-LPC is correlated with VLCFA and is a more robust measure to obtain [61]. We were able to obtain measurements of C26:0-LPC in six of the eight mouse samples we analysed and found that mutant levels were six times higher than in wildtype samples, but the difference was not significant. The numbers are listed in Appendix A.

The levels of phosphatidylinositol (3,4,5)-trisphosphate (PIP3) were increased in the *Pex3^tm1a^* mutants, specifically the molecular species stearoylarachidonoyl phosphatidylinositol triphosphate (38:4 PIP3). An increase in PIP3 has been reported to lead to changes in plasma membrane properties and decreased expression of adhesion molecules [62]. PIP2 levels influence membrane properties in the cochlea [63,64], and synaptojanin2, which dephosphorylates both PIP3 and PIP2, has been shown to be required for normal hearing [65].

The synthesis of plasmalogens and plasmanyl lipids (ether-linked lipids) starts in the peroxisomes, and decreased levels of these lipids are linked to higher membrane lipid fluidity, increased sensitivity to reactive oxygen species, and neurodegenerative disorders [66]. The inactivation of ether lipid biosynthesis has been shown to cause male infertility and eye abnormalities [67]; both of these impairments were observed in Pex3-deficient mice [18]. *Pex3* mutant mice have also been reported to show a decreased circulating cholesterol level [18], although we did not observe this feature in inner ear samples. Plasmalogens are normally enriched in lipid rafts, which are also rich in cholesterol and sphingomyelin and are thought to be important in organising domains of activity in plasma membranes [68,69]. The synaptic function of synaptosomes from the brain can be affected by a deficiency of ether lipids [70]. Thus, there may be a direct link between the reduction in ether-linked lipids, including plasmalogens, and our observations of abnormal synaptic organisation of inner hair cells. Our finding of reduced levels of plasmalogens and other ether-linked lipids and increased levels of PIP3 in the inner ear in *Pex3* mutant mice supports a key role for specific lipids in progressive hearing loss.

## Figures and Tables

**Figure 1 cells-11-03206-f001:**
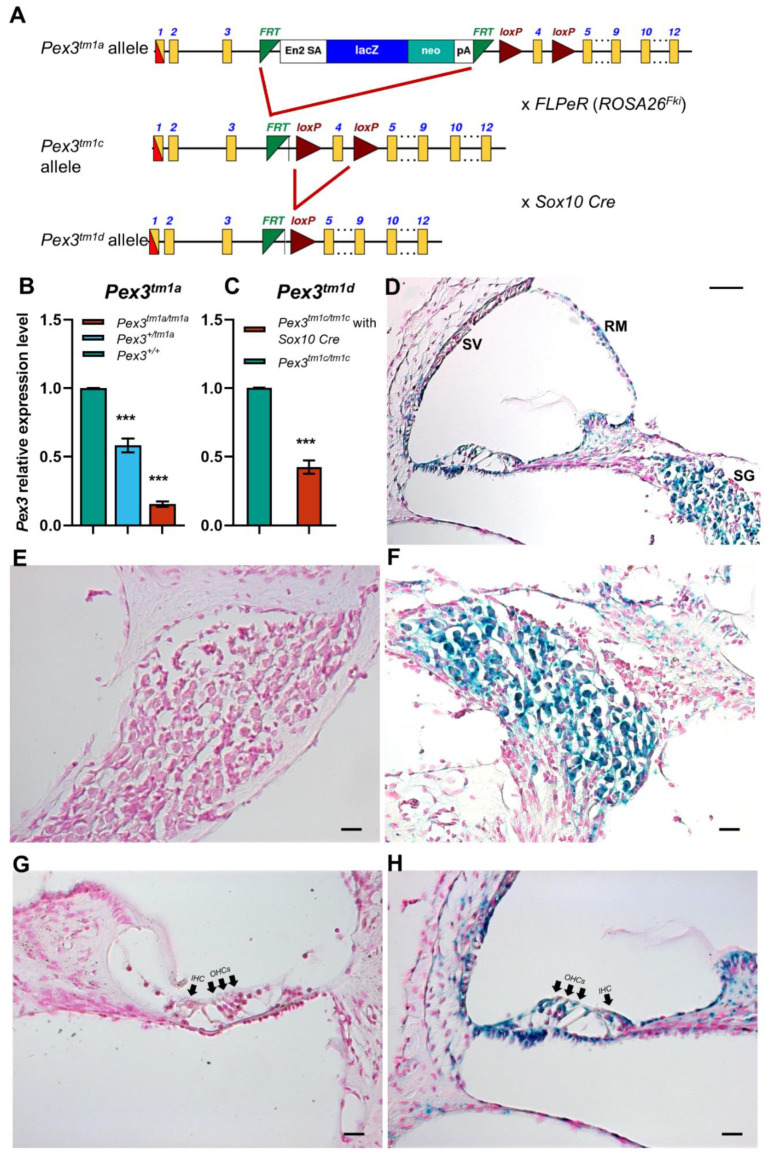
(**A**) The design of the *Pex3^tm1a^* allele is shown at the top. Exposure to Flp recombinase from the *FLPeR* allele deleted the mutagenic cassette, restoring transcription in the *Pex3^tm1c^* mutant. Exposure to Cre recombinase from the *Sox10Cre* allele deleted exon 4, also leading to a frameshift mutation in the *Pex3^tm1d^* allele. Yellow boxes, exons. Green triangles, *FRT* sites. Red triangles, *LoxP* sites. The black arrows, in intron 3 of the *Pex3* gene, represent the Pex3 forward and reverse primers used for genotyping. The red arrow is the CasR1 primer, used in combination with the Pex3 forward primer to detect the presence of the tm1a allele, and the purple arrows are the primers used to detect the neomycin resistance gene within the tm1a allele. (**B**) qPCR of brain tissues from 4-week-old mice shows reduced mRNA levels of *Pex3* expression in *Pex3^tm1a^* homozygotes (16%, red) compared with wildtype littermate controls (green), and heterozygotes have intermediate levels of transcript (58%, blue). The qPCR probe detected sequence downstream of the cassette, from exons 10–11. Data plotted as mean ± standard deviation. ***, *p* ≤ 0.001, Mann–Whitney rank sum test, *n* = 7 for homozygotes and wildtypes and *n* = 6 for heterozygotes. (**C**) qPCR of cochlear tissues from 4-week-old *Pex3^tm1d^* mutant mice shows that *Pex3* transcription level was decreased to approximately 42% in the mutants compared with the *Pex3^tm1c^* control level. Data plotted as mean ± standard deviation. ***, *p* = 0.0002. Welsh *t*-test (normal distribution, but not equal variance). *n* = 4 for each genotype. (**D**–**H**) The reporter gene *LacZ* was used to reveal the normal expression pattern of *Pex3*, shown in blue, in the spiral ganglion (**D**,**F**) and cell types around the lining of the cochlear duct (**D**,**H**) in heterozygous mice aged 2 weeks old. No blue label was detected in wildtype littermates that did not contain the *LacZ* reporter allele (**E**,**G**). IHC, inner hair cells. OHC, outer hair cells. SV, stria vascularis. SG, spiral ganglion. RM, Reissner’s membrane. Scale bars, (**D**), 50 μm; (**E**–**H**), 20 μm. *n* = 6 for each genotype.

**Figure 2 cells-11-03206-f002:**
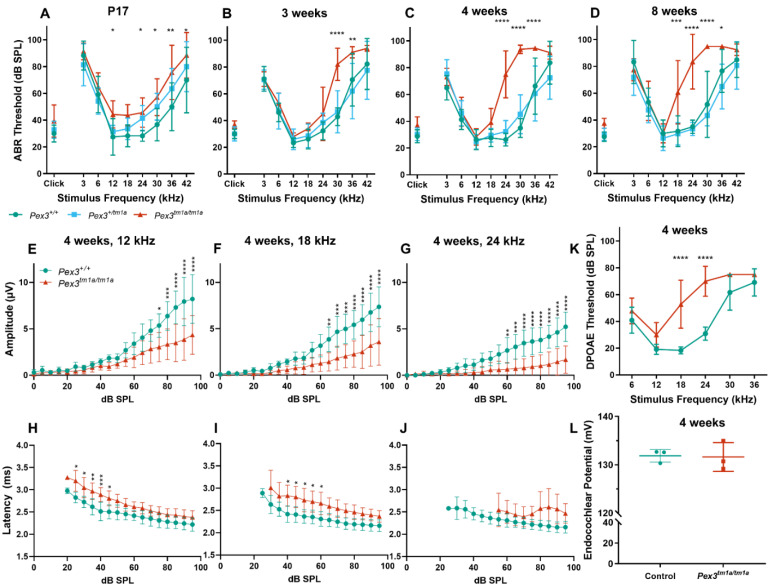
(**A**–**D**) Following maturation, ABR thresholds of *Pex3^tm1a^* homozygotes (red) showed progressive high-frequency hearing loss while heterozygotes (blue) and wildtype littermates (green) had normal thresholds. Mean ± standard deviation. For each age group, there was a significant effect of mouse genotype (P17, F(2153) = 20.02, *p* < 0.0001; three weeks, F(2189) = 19.50, *p* < 0.0001; four weeks, F(2296) = 105.3, *p* < 0.0001; eight weeks, F(2257) = 114.7, *p* < 0.0001). Significant threshold elevations for each stimulus, from Tukey’s multiple-comparisons test, between *Pex3^+/+^* and *Pex3^tm1a/tm1a^* animals, are indicated by * *p* ≤ 0.05, ** *p* ≤ 0.01, *** *p* ≤ 0.001, **** *p* ≤ 0.0001. *n* at P17 *Pex3^+/+^* = 6, *Pex3^+/tm1a^* = 7, *Pex3^tm1a/tm1a^* = 7; at three weeks *Pex3^+/+^* = 9, *Pex3^+/tm1a^* = 10, *Pex3^tm1a/tm1a^* = 5; at four weeks *Pex3^+/+^* = 4, *Pex3^+/tm1a^* = 20, *Pex3^tm1a/tm1a^* = 12; at eight weeks *Pex3^+/+^* = 3, *Pex3^+/tm1a^* = 15, *Pex3^tm1a/tm1a^* = 14. (**E**–**J**) Amplitudes and latencies of ABR wave 1 in mice aged four weeks. Homozygous mutants (red triangles) showed reduced amplitudes and extended latencies compared with the wildtype group (green circles). Input/output functions of ABR wave P1 latency and ABR wave 1 amplitude (mean ± standard deviation) are plotted vs. dB SPL, for 12 kHz (**E**,**H**), 18 kHz (**F**,**I**) and 24 kHz (**G**,**J**) stimuli. Two-way ANOVA revealed significant reductions in ABR wave amplitude (12 kHz, F (1260) = 80.51, *p* < 0.0001; 18 kHz, F (1260) = 112.2, *p* < 0.0001; 24 kHz, F (1180) = 136.2, *p* < 0.0001) and increases in P1 latency (12 kHz, F (1188) = 97.94, *p* < 0.0001; 18 kHz, F (1158) = 96.87, *p* < 0.0001; 24 kHz, F (1, 56) = 19.66, *p* < 0.0001) in homozygote *Pex3^tm1a/tm1a^* mice compared to wildtype littermate control mice. Significant changes at each dB SPL, from Sidak’s multiple-comparisons tests between *Pex3^+/+^* and *Pex3^tm1a/tm1a^* mice are indicated by * *p* < 0.05, ** *p* < 0.01, *** *p* < 0.001, **** *p* < 0.0001. (**K**) *Pex3^tm1a^* homozygotes (red) showed increased thresholds for DPOAE thresholds compared with wildtype littermate controls (green) at four weeks old. *n* = 6 wildtypes, 7 homozygous mutants. Two-way ANOVA revealed a significant effect of mouse genotype (F(1,66) = 74.65, *p* < 0.0001). Sidak’s multiple-comparisons test indicated significant elevations in DPOAE threshold at f2s of 18 kHz (*p* < 0.0001) and 24 kHz (*p* < 0.0001). (**L**) *Pex3^tm1a^* homozygotes (red) showed normal endocochlear potential in comparison with control WT littermates (green). The graph shows individual dots for each mouse and the mean (±SD) for both WT (green) and mutant (red) mice; 131.6 +/−3.0 mV in homozygous mutants; 131.9 +/−1.3 mV in littermate controls; *n* = 3 of each genotype. Data analysed by one-way ANOVA (*p* = 0.904).

**Figure 3 cells-11-03206-f003:**
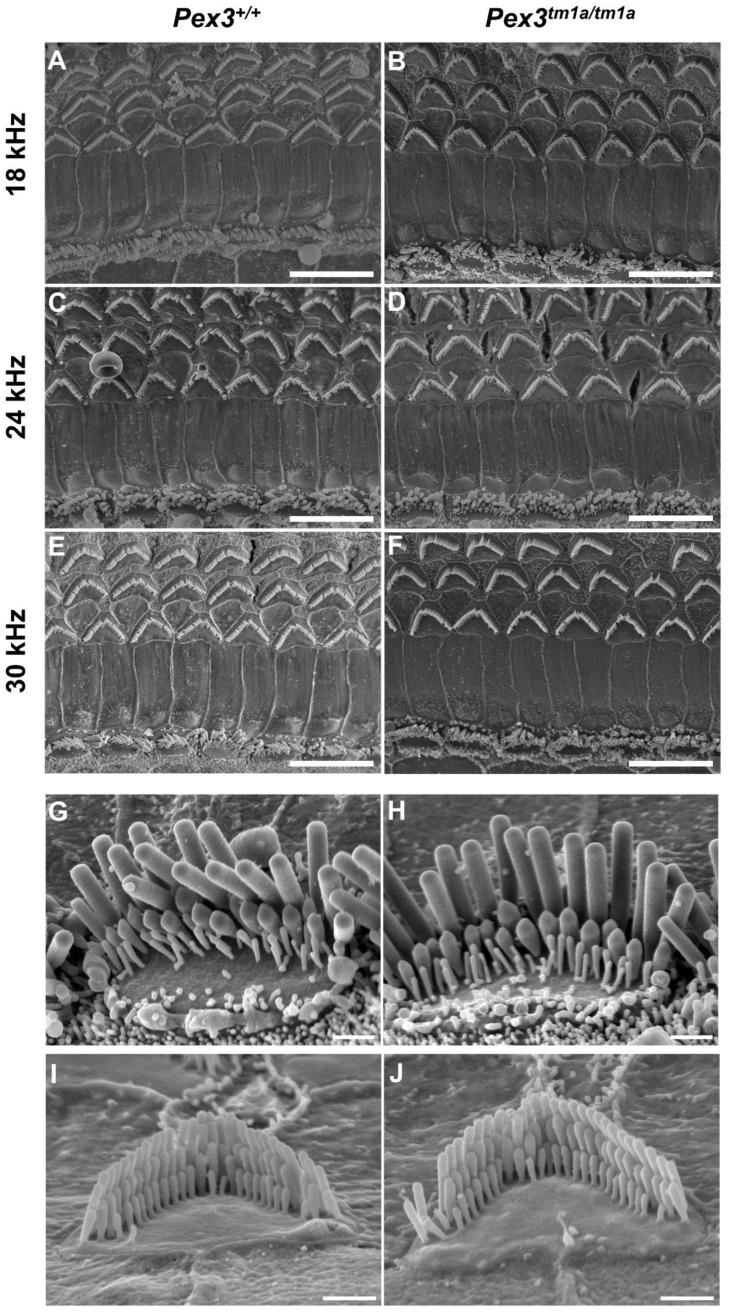
Scanning electron micrographs of the organ of Corti at four weeks old. Left column, wildtypes; right column, homozygous *Pex3^tm1a^* mutants. (**A**–**F**) low magnification images of the surface of the organ of Corti at three characteristic-frequency locations: (**A**,**B**) 18 kHz; (**C**,**D**), 24 kHz; (**E**,**F)** 30 kHz regions. (**G**–**J**), Higher magnification of stereocilia bundles. (**G**,**H**) inner hair cells from 24 kHz frequency region; (**I**,**J**), outer hair cells from 12 kHz frequency region, showing no obvious differences between *Pex3^tm1a^* mutants and controls. Occasionally, missing stereocilia bundles were observed in OHCs in both mutant and control samples. *n* = 7 for each genotype. Scale bars (**A**–**F**), 10 μm; (**G**–**J**), 1 μm.

**Figure 4 cells-11-03206-f004:**
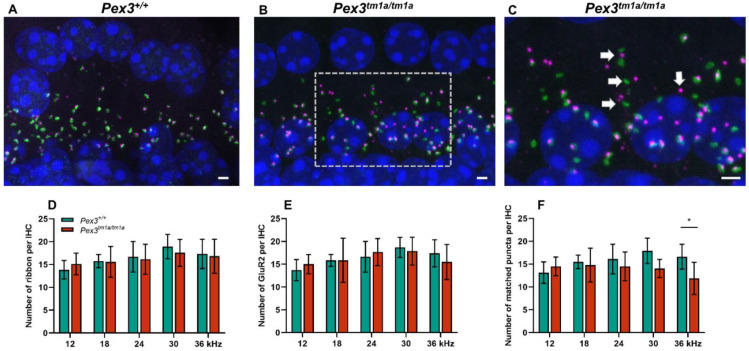
Synapses below inner hair cells shown by close apposition of the pre-synaptic ribbon (Ribeye, magenta) and the post-synaptic density (GluR2, green) in *Pex3^tm1a^* homozygotes and wildtype controls. Nuclei are labelled with DAPI, blue. Maximum projection confocal images. (**A**) Wildtype mouse at four weeks old, 30 kHz best-frequency region. (**B**) *Pex3^tm1a^* homozygous littermate. (**C**) Enlargement of box in B with white arrows pointing to separated pre- and post-synaptic components. (**D**–**F**) plots of numbers of ribbons per IHC (**D**), post-synaptic densities per IHC (**E**) and intact synapses per inner hair cell (**F**) at different cochlear regions, wildtypes in green, homozygous mutants in red. All data passed the normality and equal-variance tests. Two-way ANOVA revealed significant reductions in number of matched puncta (F (1, 50) = 7.042, *p* = 0.0106) in *Pex3^tm1a^* homozygotes in comparison with littermate controls, but not in the number of ribbons (F (1, 50) = 0.1208, *p* = 0.7296) nor GluR2 puncta (F (1, 50) = 0.008970, *p* = 0.9249). Significant changes at each frequency, from Sidak’s multiple-comparisons tests between *Pex3^+/+^* and *Pex3^tm1a/tm1a^* mice are indicated by * *p* < 0.05. Scale bar, 2 μm; *n* = 6 for each genotype. Number of synapses/IHC (±SD): *Pex3^+/+^* control mice 12 kHz 13.13 ± 2.35; 18 kHz 15.50 ± 1.49; 24 kHz 16.11 ± 3.24; 30 kHz 17.91 ± 2.78; 36 kHz 16.63 ± 2.73. *Pex3^tm1a/tm1a^* mutant mice 12 kHz 14.50 ± 2.05; 18 kHz 14.05 ± 2.00; 24 kHz 14.52 ± 3.14; 30 kHz 14.06 ± 1.98; 36 kHz 11.87 ± 3.54.

**Figure 5 cells-11-03206-f005:**
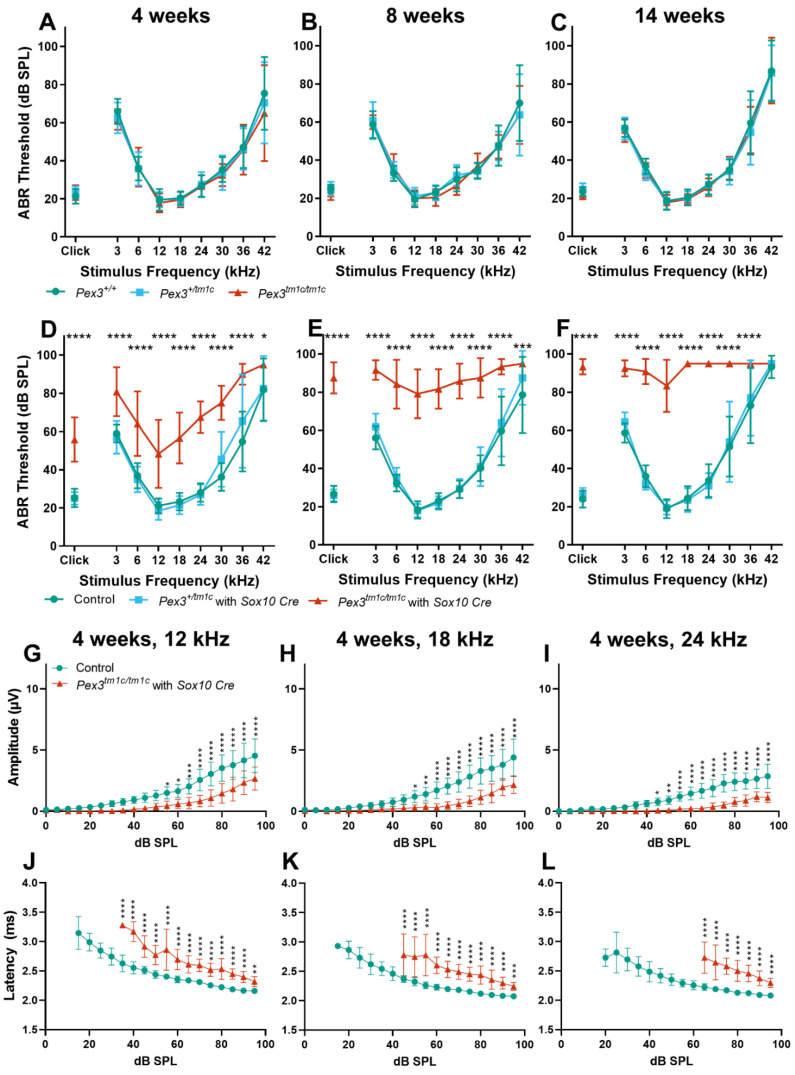
(**A**–**C**) ABR thresholds of the *Pex3^tm1c^* allele, with most of the introduced DNA cassette removed, restoring normal ABR thresholds. The same mice were tested at the three ages. *Pex3^+/+^, n* = 11; *Pex3^+/tm1c^*, *n* = 12; *Pex3^tm1c/tm1c^*, *n* = 12. Two-way ANOVA revealed no effect on thresholds of the mouse genotype (4 weeks, F(2288) = 1.170, *p* = 0.3117; 8 weeks, F(2288) = 0.4605, *p* = 0.6314; 14 weeks, F(2288) = 0.6117, *p* = 0.5431). (**D**–**F**) ABR thresholds of the *Pex3^tm1d^* mutant mice, which had exon 4 deleted in *Sox10*-expressing tissues including all cell types of the inner ear (see Figure 1A for allele details). Mice that were homozygous for the *Pex3^tm1c^* allele and that also carried the *Sox10Cre* allele (red) showed severe progressive hearing impairment at all frequencies. Two-way ANOVA revealed a significant effect on threshold of the genotype of the mouse (4 weeks, F(2306) = 184.4, *p* < 0.0001; 8 weeks F(2306) = 519.1, *p* < 0.0001; 14 weeks, F(2297) = 536.5, *p* < 0.0001). Tukey’s multiple-comparisons tests indicated threshold elevations in *Pex3^tm1c/tm1c^* mice carrying the *Sox10Cre* allele vs. controls in response to stimuli, indicated by * *p* < 0.05, 0.01, *** *p* < 0.001, **** *p* < 0.0001. At 4 weeks: controls *n* = 21, *Pex3^+/tm1c^* with *Sox10Cre n* = 10, *Pex3^tm1a/tm1a^ n* = 6; at 8 weeks: controls *n* = 21, *Pex3^+/tm1c^* with *Sox10Cre n* = 10, *Pex3^tm1a/tm1a^ n* = 6; at 14 weeks: controls *n* = 20, *Pex3^+/tm1c^* with *Sox10Cre n* = 10, *Pex3^tm1a/tm1a^ n* = 6. The control groups included the following genotypes: *Pex3^+/+^*, *Pex3^+/tm1c^*, *Pex3^tm1c/tm1c^*, and *Pex3^+/+^* with *Sox10Cre*. (**G**–**L**) Amplitudes and latencies of ABR wave 1 in mice aged four weeks. Homozygous *Pex3^tm1c/tm1c^* mutants carrying the *Sox10Cre* allele (red triangles) showed reduced amplitudes and extended latencies compared with the control group (green circles). Input/output functions of ABR wave P1 latency and ABR wave 1 amplitude (mean ± standard deviation) are plotted vs. dB SPL, for 12 kHz (**G**,**J**), 18 kHz (**H**,**K**) and 24 kHz (**I**,**L**) stimuli. Two-way ANOVA revealed significant reductions in ABR wave amplitude (12 kHz, F (1, 500) = 209.7, *p* < 0.0001; 18 kHz, F (1, 500) = 226.1, *p* < 0.0001; 24 kHz, F (1, 500) = 333.9, *p* < 0.0001) and increases in P1 latency (12 kHz, F (1, 308) = 718.0, *p* < 0.0001; 18 kHz, F (1, 261) = 634.1, *p* < 0.0001; 24 kHz, F (1, 170) = 553.9, *p* < 0.0001) in homozygote mice compared to wildtype controls. Significant changes at each dB SPL, from Sidak’s multiple-comparisons tests between *Pex3^+/+^* and *Pex3^tm1c/tm1a^* with *Sox10-Cre*, are indicated by * *p* < 0.05, ** *p* < 0.01, *** *p* < 0.001, **** *p* < 0.0001.

**Figure 6 cells-11-03206-f006:**
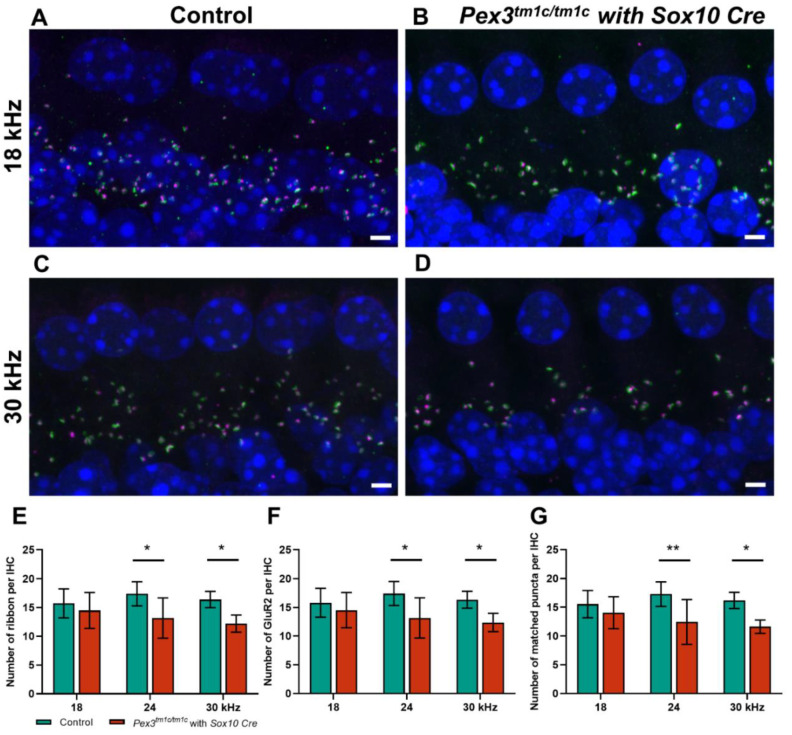
(**A**–**D**) Maximum projection confocal images of *Pex3^tm1d^* homozygotes (**B**,**D**) and their littermate controls (**A**,**C**) aged four weeks at 18 and 24 kHz best-frequency locations. Pre-synaptic ribbon (Ribeye, magenta); post-synaptic density (GluR2, green); nuclei (DAPI, blue). (**E**–**G**) numbers of intact synapses per inner hair cell at different cochlear regions; controls in green; homozygous mutants in red. All synaptic data passed the normality and equal-variance tests. Two-way ANOVA revealed significant reductions in number of matched puncta (F (1, 27) = 18.53, *p* = 0.0002), ribbons (F (1, 27) = 14.10, *p* = 0.0008) and GluR2 puncta (F (1, 27) = 13.70, *p* = 0.0010) in *Pex3^tm1d^* homozygotes in comparison with littermate controls. Significant changes at each frequency, from Sidak’s multiple-comparisons tests between *Pex3^+/+^* and *Pex3^tm1d/tm1d^* mice, are indicated by * *p* < 0.05 and ** *p* < 0.01. Scale bar, 2 μm; *n*, *Pex3^+/+^* = 6 and *Pex3^tm1d/tm1d^* = 5. Number of synapses/IHC (±SD): Control 18 kHz 15.55 ± 2.37; 24 kHz 17.26 ± 2.39; 30 kHz 16.19 ± 1.40. *Pex3^tm1d/tm1d^* mutants 18 kHz 14.05 ± 2.79; 24 kHz 12.45 ± 3.90; 30 kHz 11.63 ± 1.15.

**Figure 7 cells-11-03206-f007:**
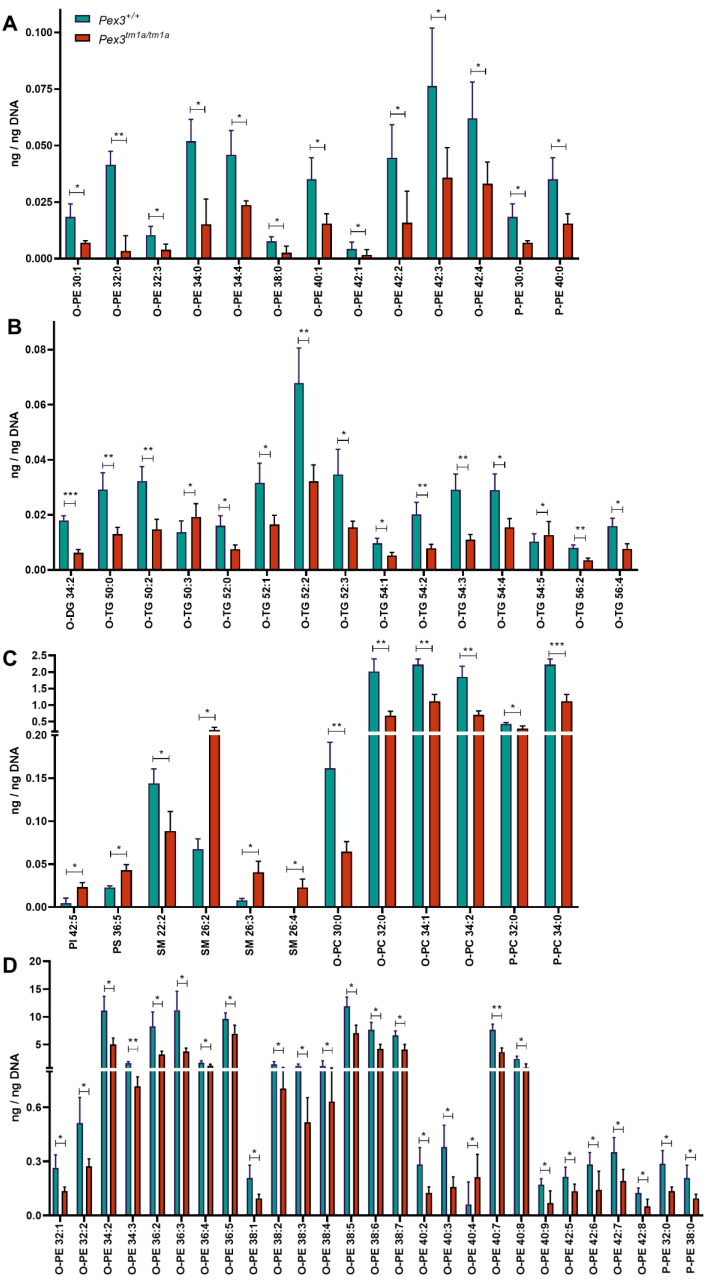
Lipidomics analysis of whole inner ears. (**A**–**D**) The mean of the four *Pex3^tm1a^* mutant homozygote inner ear samples was compared with the mean of their littermate wildtype controls. Out of 671 lipid molecular species analysed, only the quantities (ng/ng DNA) of single molecular species showing a significant decrease or increase are plotted. *t*-test, * *p* < 0.05, ** *p* < 0.01, *** *p* < 0.001. Abbreviations of lipid subclasses are given in Table 1. Appendix A presents the quantities per mouse sample of all 671 lipid molecular species examined.

**Figure 8 cells-11-03206-f008:**
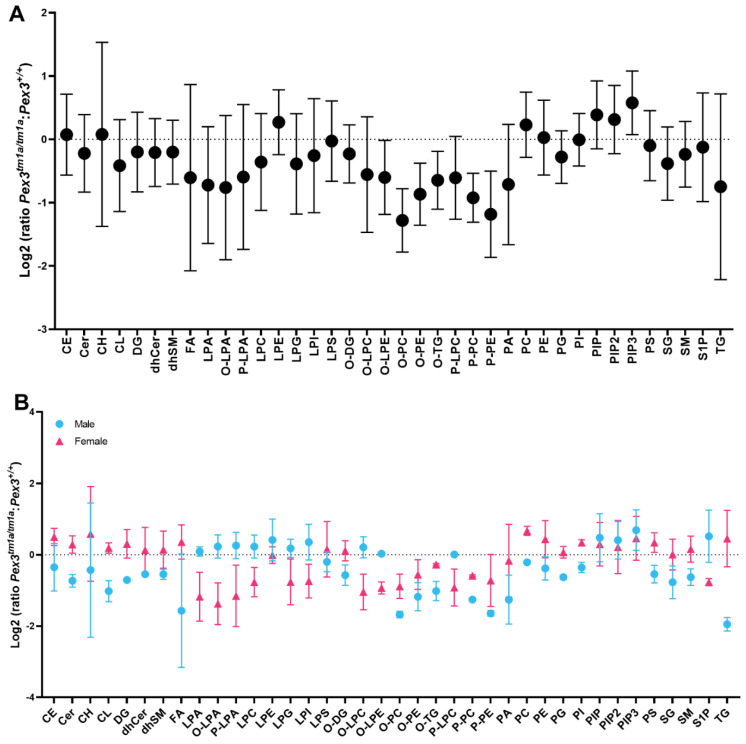
Lipidomics analysis of whole inner ears. Amounts of each lipid subclass in each homozygous *Pex3^tm1a^* mutant were compared with its sex-matched wildtype littermate control, and the ratios of each pair were averaged and plotted as fold change with 95% confidence intervals. If the ratio is >0, then the lipid level is increased in the mutants, and if it is <0, then the lipid level is decreased in the mutants. (**A**) Log_2_ fold change analysis of littermates showed a reduction in the mutant inner ear of plasmalogens (P-PC, P-PE) and other lipids (O-LPE, O-PC, O-PE, O-TG) indicated by the lack of overlap between the standard deviation bars with the baseline (dotted line). PIP3 ratios were increased in mutants. *n* = 4 pairs. (**B**) Lipid ratios from male and female pairs are plotted separately, showing reductions in ether lipids including plasmalogens in both sexes. Females have increased TG and FA in comparison to males. *n* = 2 male pairs, 2 female pairs. Abbreviations of lipid subclasses are given in Table 1. Appendix A presents the quantities per mouse sample of all 671 lipid molecular species examined.

**Table 1 cells-11-03206-t001:** Main subclasses of lipids detected with their abbreviations.

Subclass of Lipid	Short Form	Class of Lipid	Subclass
Phosphatidic acid	PA	Glycerophospholipids	
Phosphatidylcholine	PC	Glycerophospholipids	
Phosphatidylethanolamine	PE	Glycerophospholipids	
Phosphatidylinositol	PI	Glycerophospholipids	
Phosphatidylserine	PS	Glycerophospholipids	
Phosphatidylglycerol	PG	Glycerophospholipids	
Cardiolipin	CL	Glycerophospholipids	
Alkyl-acylphosphatidylcholine	O-PC	Glycerophospholipids	
Alkenyl-acylphosphatidylcholine	P-PC	Glycerophospholipids	Plasmalogen
Alkyl-acylphosphatidylethanolamine	O-PE	Glycerophospholipids	
Alkenyl-acylphosphatidlylethanolamine	P-PE	Glycerophospholipids	Plasmalogen
Lysophosphatidylcholine	LPC	Glycerophospholipids	
Lysophosphatidylelthanolamine	LPE	Glycerophospholipids	
Lysophosphatidylserine	LPS	Glycerophospholipids	
Lysophosphatidic acid	LPA	Glycerophospholipids	
Lysophosphatidylinositol	LPI	Glycerophospholipids	
Lysophosphatidylglycerol	LPG	Glycerophospholipids	
Alkyl-Lysophosphatidylcholine	O-LPC	Glycerophospholipids	
Alkyl-Lysophosphatidylethanolamine	O-LPE	Glycerophospholipids	
Alkyl-Lysophosphatidic acid	O-LPA	Glycerophospholipids	
Alkenyl-Lysophosphatidic acid	P-LPA	Glycerophospholipids	
Alkenyl-Lysophosphatidylcholine	P-LPC	Glycerophospholipids	
Phosphatidylinositol phosphate	PIP	Glycerophospholipids	
Phosphatidylinositol bisphosphate	PIP2	Glycerophospholipids	
Phosphatidylinositol trisphosphate	PIP3	Glycerophospholipids	
Diacylglycerol	DG	Glycerolipids	
Triacylglycerol	TG	Glycerolipids	
Alkyl−acylglycerol	O-DG	Glycerolipids	
Alkyl−triacylglycerol	O-TG	Glycerolipids	
Sphingosine-1-phosphate	S1P	Sphingolipids	
Sphingosine	SG	Sphingolipids	
Dihydroceramide	dhCer	Sphingolipids	
Ceramides	Cer	Sphingolipids	
Sphingomyelin	SM	Sphingolipids	
Dihydrosphingomyelin	dhSM	Sphingolipids	
Cholesterol Ester	CE	Sterols	
Cholesterol	CH	Sterols	
Free fatty acids	FA	Fatty acids	

## Data Availability

Lipidomics data are presented in Appendix A. Additional data are presented in the text. Further information is available upon request to the corresponding author.

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
