# Peer review of "The Effect of a Pex3 Mutation on Hearing and Lipid Content of the Inner Ear"

_cells, 2022, doi:10.3390/cells11203206_

Round 1

Reviewer 1 Report

The authors have the solid beginnings of a manuscript. Nevertheless, the biochemical analyses of peroxisomal metabolites leaves something to be desired. VLCFA (very-long-chain fatty acid) levels have to be measured to make this a complete manuscript. They are a hallmark of peroxisomal metabolite analyses. It would be useful to measure specific plasmalogen species as well; however, that is more optional given that relative levels of groups of plasmalogens were reported. Figure 6 was confusing in that there were conflicting results for the SM lipid species (i.e., data using the 22:2, 26:2, 26:3, and 26:4 standards gave conflicting results (see Panel C)). Figure 7 Panel B would be better off given in table format since the order of the blue and pink dots seems to change across the figure and are so crowded that it is confusing to the reader. The authors should be lauded for being forthcoming about the fact that the CKO animals were mosaic given incomplete recombination occurring at the Pex3 loxP sites. At the same time, this fact should be discussed further so the readers understand the implications of this mosaicism on the phenotypes of interest. Minor edits: line 34 "peroxisome membrane proteins" instead of peroxisome matrix proteins I would suggest using updated reference throughout the manuscript whenever possible.

Overall, this is a reasonable manuscript of interest to the field that would benefit from the inclusion of VLCFA data and perhaps the measurement of specific plasmalogens species.

Author Response

The authors have the solid beginnings of a manuscript. Nevertheless, the biochemical analyses of peroxisomal metabolites leaves something to be desired. VLCFA (very-long-chain fatty acid) levels have to be measured to make this a complete manuscript. They are a hallmark of peroxisomal metabolite analyses.

We were able to identify long chain fatty acids up to C24:0 but not longer than this in the very small samples we obtained from the two inner ears of each mouse. Raised levels of very long chain fatty acids (VLCFA; C26:0 and C26:0/C22:0 ratio) in plasma have traditionally been used as diagnostic for patients with peroxisomal disorders (eg Klouwer et al 2015; Moser et al 1984; Stradomska et al 2020), but recently it has been suggested that C26:0-LPC is correlated with VLCFA and is a more robust measure to obtain (Jaspers et al 2020). We were able to obtain measurements of C26:0-LPC in six of the eight mouse samples we analysed and found that mutant levels were six times higher than in wild type samples, but the difference was not significant. The numbers are listed in a new table, Supplementary Table 1. We have added new text to the lipidomics discussion to describe our findings.

It would be useful to measure specific plasmalogen species as well; however, that is more optional given that relative levels of groups of plasmalogens were reported.

Figure 7A (old figure 6A) presents the results from plasmalogen molecular species that showed significant differences between mutant and control samples. Other plasmalogen species that were not significantly different are not presented in the figure but we have now added a supplementary table listing all of the 671 lipids we examined including those with no significant difference.

Figure 6 was confusing in that there were conflicting results for the SM lipid species (i.e., data using the 22:2, 26:2, 26:3, and 26:4 standards gave conflicting results (see Panel C)).

The results plotted in Figure 6 are correct. We found significantly increased levels of sphingomyelin lipids 22:2, 26:2, 26:3 and 26:4 in the mutant samples. It is not clear why the reviewer considers this to be a conflicting finding.

Figure 7 Panel B would be better off given in table format since the order of the blue and pink dots seems to change across the figure and are so crowded that it is confusing to the reader.

We have redrawn this panel (new figure 8B) to link the male and female of each pair more clearly.

The authors should be lauded for being forthcoming about the fact that the CKO animals were mosaic given incomplete recombination occurring at the Pex3 loxP sites. At the same time, this fact should be discussed further so the readers understand the implications of this mosaicism on the phenotypes of interest.

We have changed the description from lines 432-436 (original line numbers) to read as follows, as the reviewer suggests.

“Sequencing of cDNA from the inner ear indicated that some sequence from exon 4 was present in the samples. This suggested that the Cre recombinase activity was not fully efficient in the inner ear tissues tested leading to mosaicism, with some cells still carrying the Pex3tm1c allele containing exon 4. To produce a complete knockout, the Cre recombinase needs to be highly efficient as it needs to cut exon 4 from both copies of the Pex3 gene in every cell. Despite the observed retention of some copies of exon 4 revealed by our sequencing, the overall effect of the Sox10-driven Cre recombinase was enough to lead to raised ABR thresholds across all frequencies.”

Minor edits: line 34 "peroxisome membrane proteins" instead of peroxisome matrix proteins

We have changed this as suggested by the reviewer.

I would suggest using updated reference throughout the manuscript whenever possible.

We have added further references where appropriate.

Overall, this is a reasonable manuscript of interest to the field that would benefit from the inclusion of VLCFA data and perhaps the measurement of specific plasmalogens species.

Specific plasmalogens are listed in the new supplementary table.

Reviewer 2 Report

This manuscript describes two Pex3 mouse mutants: one Pex3 knockdown with lac-z reporter, and one conditional Pex3 knockout using Sox10Cre investigating the potential role of Pex3 in the cochlea. Evaluations were performed in the Pex3 knockdown, including functional hearing analysis, RNA analysis, measurements of the endocochlear potential, histology, scanning electron microscopy, and lipidomics. In both mouse models, there was some characterization of synapses.

Major:

The authors demonstrated that Pex3 expression was knocked down to 16% of normal levels in the Pex3tm1a model which had mild high frequency hearing loss. In contrast, Pex3 expression was reduced to 42% in the conditional knockout model and these mice had severe hearing loss across all frequencies. If Pex3 insufficiency is driving the phenotype, why are the differences more severe in the model with less knockdown?

The conditional knockout using Sox10Cre is not characterized beyond examining synapses in 3 areas along the cochlear duct which is surprising given its severe phenotype. As Sox10 is expressed in most cells in the cochlea including cells of the stria vascularis, characterizing the morphology and function of the stria by measuring endocochlear potential seems very relevant. It is important to point out that Sox10 is not expressed in the sensory hair cells or spiral ganglion neurons making the observed and limited disruption of synapses at 24 and 30 kHz harder to explain especially considering that the synapses were normal at 18 kHz.

Other major comments

1.      To help visualise lac-z expression in the entire cochlea, a zoomed-out image of the entire mid-modiolar section should be included. While the authors mention that there is expression in the stria vascularis, there are no images of the stria. Also, to improve the resolution and cellular detection, co-labelling would identify which cell types in the stria express Pex3. This approach would also resolve the expression of Pex3 in the organ of Corti with co-labelling with markers for hair cells and supporting cells and in the spiral ganglion with markers for neurons and glia.

2.      As you already have click ABR data, an analysis of Peak Height and Latency could be performed. Peak 1 height and subsequent peak latencies would be particularly useful given the synapse pathology.

3.      Line 320 (and Fig. 2 legend) describes ‘near-normal’ hearing indicative of normal development in the Pex3tm1a mouse. However, Fig. 2A shows a significant difference between homozygote and WT ABRs.

4.      Please check the variability shown in Figure 2 for the Pex3tm1a ABR timepoints, with hearing improving in the Pex3tm1a mouse. Specifically, significance was observed at 12, 24, 30, 36 and 42 kHz for P17, but only 30 and 36 for 3 weeks. Then 24, 30 and 36 for 4 weeks and 18-36 at 8 weeks.

5.      Line 385 says ‘12 and 18 kHz where mutant thresholds were normal’, however in Figure 2A there is a significant difference at 12 kHz and a trend for 18. Figure 2C has a similar trend, although the mutant hearing has improved at 12 kHz. Can you overcome this ambiguity by quantifying synapses at the lower frequencies where there is clearly no difference?

6.      Figure 5 shows differences between matched synapse number at 24 and 30 kHz, but not 18 kHz. This seems strange given the huge difference between ABRs at these frequencies. Why is the hearing of Pex3 conditional knockouts so bad if synapses are not affected? This mouse model needs to be better characterized to identify the cellular phenotype resulting in severe hearing loss.

7.      Line 443: “Sox-10 expressing tissues including all cell types of the inner ear”; however, Sox 10 is not expressed in hair cells or spiral ganglion neurons and thus this should be made clear in the manuscript.

8.      Line 509 says the Pex3 gene was effectively knocked out using a conditional approach, but this contradicts line 432 which states that expression was reduced but not eliminated (i.e. Pex3 transcription reduced to 42%).

9.      Line 510 says “These findings suggest a graded response of auditory function to varying levels of Pex3 transcription, with the basal turn showing the greatest sensitivity to Pex3 levels” but this is not supported by the data presented in 5D-F, nor the Pex3 expression provided for both models.

1.  It was difficult to follow the statistical analysis described in the methods sections, which does not always match the figure legends. In some cases, a one-way ANOVA has been used but this cannot provide the specific p values/comparisons that have been given in the text. The significance described on line 407 does not appear to be accurate given the numbers provided in the figure legend (408-409) and significance is given for 4F, but not for 4D or E.

Minor:

1.      A better description of PEX mutations symptoms in the abstract and introduction could be more informative.

2.      The abstract states “the earliest structural defect was disruption of synapses”. This suggests that there were other defects; however, the remaining structures in the ear are described as normal.

3.      In general, figure legends contain typos and can be difficult to follow.

4.      Line 530 says ‘improving lipid metabolism systemically may not be effective in reversing hearing loss’ Since, data indicate that in Pex3tm1a hearing loss is progressive, could synapse loss be prevented with treatment?

5.      Lipidomic data is largely ignored in the discussion. Further explanation of the findings would be useful.

Author Response

This manuscript describes two Pex3 mouse mutants: one Pex3 knockdown with lac-z reporter, and one conditional Pex3 knockout using Sox10Cre investigating the potential role of Pex3 in the cochlea. Evaluations were performed in the Pex3 knockdown, including functional hearing analysis, RNA analysis, measurements of the endocochlear potential, histology, scanning electron microscopy, and lipidomics. In both mouse models, there was some characterization of synapses.

Major:

The authors demonstrated that Pex3 expression was knocked down to 16% of normal levels in the Pex3tm1a model which had mild high frequency hearing loss. In contrast, Pex3 expression was reduced to 42% in the conditional knockout model and these mice had severe hearing loss across all frequencies. If Pex3 insufficiency is driving the phenotype, why are the differences more severe in the model with less knockdown?

The expression was an examination of the amount of mRNA generated by each allele. Protein transcribed from the Pex3tm1a allele would likely be qualitatively normal because no exons are deleted. However, the conditional deletion removed a critical exon so we would expect the impact upon any resulting protein to be driven by the missing sequence and hence more severe. We do not suggest that Pex3 mRNA insufficiency drives the phenotype in the conditional allele. We have added some text to discuss further the mechanism underlying the Pex3tm1d allele (see below), which is in contrast to that of the Pex3tm1a allele described earlier in the paper.

“Sequencing of cDNA from the inner ear indicated that some sequence from exon 4 was present in the samples. This suggested that the Cre recombinase activity was not fully efficient in the inner ear tissues tested leading to mosaicism, with some cells still carrying the Pex3tm1c allele containing exon 4. To produce a complete knockout, the Cre recombinase needs to be highly efficient as it needs to cut exon 4 from both copies of the Pex3 gene in every cell. Despite the observed retention of some copies of exon 4 revealed by our sequencing, the overall effect of the Sox10-driven Cre recombinase was enough to lead to raised ABR thresholds across all frequencies.”

The conditional knockout using Sox10Cre is not characterized beyond examining synapses in 3 areas along the cochlear duct which is surprising given its severe phenotype. As Sox10 is expressed in most cells in the cochlea including cells of the stria vascularis, characterizing the morphology and function of the stria by measuring endocochlear potential seems very relevant. It is important to point out that Sox10 is not expressed in the sensory hair cells or spiral ganglion neurons making the observed and limited disruption of synapses at 24 and 30 kHz harder to explain especially considering that the synapses were normal at 18 kHz.

We focussed our structural analysis on synapses in the conditional allele because this was the key defect we detected in the Pex3tm1a allele. We also show progressive increase in ABR thresholds in the conditional allele. However, regarding Sox10 expression, it is expressed in all cells of the otic placode early in development at embryonic day 8.5 to 9 as well as in migrating neural crest cells, so would drive deletion of the floxed allele of Pex3 in all cells of the inner ear including hair cells and spiral ganglion cells (Breuskin et al., 2009; Wakaoka et al., 2013; Watanabe et al., 2000). Further details and references are now included at lines 91-92 (original line numbers).

Other major comments

  1. To help visualise lac-z expression in the entire cochlea, a zoomed-out image of the entire mid-modiolar section should be included. While the authors mention that there is expression in the stria vascularis, there are no images of the stria. Also, to improve the resolution and cellular detection, co-labelling would identify which cell types in the stria express Pex3. This approach would also resolve the expression of Pex3 in the organ of Corti with co-labelling with markers for hair cells and supporting cells and in the spiral ganglion with markers for neurons and glia.

We have included a zoomed-out image of the cochlear duct showing broad expression of the reporter as requested.  See revised Figure 1.

As most cells around the cochlear duct and spiral ganglion express Pex3 judging by the extensive LacZ reporter labelling, we have not tried to co-label with other markers.

  1. As you already have click ABR data, an analysis of Peak Height and Latency could be performed. Peak 1 height and subsequent peak latencies would be particularly useful given the synapse pathology.

This is a useful suggestion and we have now included details of the amplitude and latency of the ABR wave 1 in response to clicks, as suggested by the reviewer.  The reduced amplitude and prolonged latency of this wave supports a synaptic defect even at frequencies where thresholds are normal and the reduced synaptic numbers are not yet observed. See revised figures 2 and 5. We have added text to the results and discussion to describe these new analyses and the implications.

  1. Line 320 (and Fig. 2 legend) describes ‘near-normal’ hearing indicative of normal development in the Pex3tm1a mouse. However, Fig. 2A shows a significant difference between homozygote and WT ABRs.

Yes the reviewer is correct, there is a statistically-significant difference in ABR thresholds of mutants at P17.  There is always variability in thresholds at these young ages and the difference is not a large effect size. We have changed the text to say “only slightly raised ABR thresholds at that age, suggesting largely normal development” instead of “near-normal”. We have also added the following text in the results:

“All three genotypes continue to develop more mature ABR thresholds between P17 and P21, as is often seen, but the slightly raised thresholds in the mutants at P17 may reflect a delay in this maturation process which may correspond to the developmental delay reported in children with Zellweger syndrome.”

  1. Please check the variability shown in Figure 2 for the Pex3tm1aABR timepoints, with hearing improving in the Pex3tm1a mouse. Specifically, significance was observed at 12, 24, 30, 36 and 42 kHz for P17, but only 30 and 36 for 3 weeks. Then 24, 30 and 36 for 4 weeks and 18-36 at 8 weeks.

We have checked the statistics and the results indicated by asterisks are correct. There is always more variability in ABR thresholds at young ages such as P17 because hearing is developing rapidly and pups that are slightly faster or slower in developing can show wide differences in thresholds. They tend to catch up and show more mature thresholds by around three weeks old. The P17 data do not neatly follow the progressive inclusion of lower frequencies with age as the reviewer points out, but we suggest that the P17 results are more likely due to developmental delay than early deterioration of function. We have added some text to point out the possible developmental delay, which may correspond to the developmental delay reported in Zellweger syndrome (see previous point).

  1. Line 385 says ‘12 and 18 kHz where mutant thresholds were normal’, however in Figure 2A there is a significant difference at 12 kHz and a trend for 18. Figure 2C has a similar trend, although the mutant hearing has improved at 12 kHz. Can you overcome this ambiguity by quantifying synapses at the lower frequencies where there is clearly no difference?

As we mentioned above, the ABR thresholds are usually more variable over the few days after the onset of responses, including at P17 (Fig 2A), while thresholds are maturing. We analysed synapses at P28, and given the variability at P17 as discussed above, we think that analysing synapses at the 12kHz location at P28 is appropriate.

  1. Figure 5 shows differences between matched synapse number at 24 and 30 kHz, but not 18 kHz. This seems strange given the huge difference between ABRs at these frequencies. Why is the hearing of Pex3 conditional knockouts so bad if synapses are not affected? This mouse model needs to be better characterized to identify the cellular phenotype resulting in severe hearing loss.

Yes, it is true that there is a large difference in thresholds at 18kHz but no detectable difference in synapse numbers.  We suggest that dysfunction usually precedes detectable structural changes, which would explain our findings. We have now analysed the ABR waveforms in more detail and report reduced amplitudes and increased latencies of wave 1, indicating abnormal function at the inner hair cell/spiral ganglion neuron interface, at 12, 18 and 24 kHz (see revised figures 2 and 5). There are other studies that could be carried out but these data are what we have at the moment.

  1. Line 443: “Sox-10 expressing tissues including all cell types of the inner ear”; however, Sox 10 is not expressed in hair cells or spiral ganglion neurons and thus this should be made clear in the manuscript.

Please see note above about Sox10 expression in all cells contributing to the inner ear at a much earlier stage of development, in the otic placode.

  1. Line 509 says the Pex3 gene was effectively knocked out using a conditional approach, but this contradicts line 432 which states that expression was reduced but not eliminated (i.e. Pex3 transcription reduced to 42%).

We have clarified the points made by the reviewer by specifying mRNA on line 432 and Pex3 protein on line 509 (original line numbers). Furthermore, we have added some text to discuss further the different mechanisms underlying the two alleles (see lines 432-436, original submission line numbers).

  1. Line 510 says “These findings suggest a graded response of auditory function to varying levels of Pex3 transcription, with the basal turn showing the greatest sensitivity to Pex3 levels” but this is not supported by the data presented in 5D-F, nor the Pex3 expression provided for both models.

We have reworded the paragraph to clarify that the comment about the increased sensitivity of the basal turn refers to the Pex3tm1a allele.  

  1. It was difficult to follow the statistical analysis described in the methods sections, which does not always match the figure legends. In some cases, a one-way ANOVA has been used but this cannot provide the specific p values/comparisons that have been given in the text. The significance described on line 407 does not appear to be accurate given the numbers provided in the figure legend (408-409) and significance is given for 4F, but not for 4D or E.

Thank you for pointing out this inconsistency.  We have repeated the statistical analysis using two-way ANOVA and corrected figure 4 and new figure 6 and their legends accordingly.

Minor:

  1. A better description of PEX mutations symptoms in the abstract and introduction could be more informative.

We have added further details to the introduction from line 36 as follows:

“Mutations in PEX genes lead to a spectrum of diseases called peroxisome biogenesis disorders (PBD), the most severe of which is Zellweger syndrome which includes hypotonia, renal and liver defects, seizures, developmental delay, and often leads to early childhood death (Waterham and Ebberink 2012). Depending on the specific mutation, individuals carrying PEX mutations can show milder effects but hearing loss is a frequent feature of people on the Zellweger spectrum (Klouwer et al 2015; Lee et al 2022).”

  1. The abstract states “the earliest structural defect was disruption of synapses”. This suggests that there were other defects; however, the remaining structures in the ear are described as normal.

We have changed this to “The only structural defect seen at 4 weeks old …”

  1. In general, figure legends contain typos and can be difficult to follow.

We have checked carefully for typos and apologise that the legends are difficult to read.  The statistical details need to be included in the legends.

  1. Line 530 says ‘improving lipid metabolism systemically may not be effective in reversing hearing loss’ Since, data indicate that in Pex3tm1ahearing loss is progressive, could synapse loss be prevented with treatment?

There is currently no treatment available but it is obvious that a treatment would be desirable.  The point we make here is that it would probably not be enough to correct the systemic lipid profile to rescue the hearing of people with PEX3 mutations, because PEX3 is required locally in the inner ear for normal function. We have added some text to the discussion to present the context more clearly: “Devising a treatment to improve lipid balance in the circulation could be simpler than developing site-specific treatments.”

  1. Lipidomic data is largely ignored in the discussion. Further explanation of the findings would be useful.

This is a surprising comment, as nearly half of the discussion (original lines 542-575) is about the lipidomics findings. We have now added a new paragraph to the discussion that relates the increase in VLCFA and C26:0-LPC levels observed in peroxisomal disorders in humans to our own observations.

Reviewer 3 Report

“The effect of a Pex3 mutation on hearing and lipid content of the inner ear” is a very well written and comprehensive article.

There are few questions:

Q1: Zellweger spectrum disorder sometimes is associated with mixed hearing loss. Were those aspect of the hearing loss evaluated?

Q2: Were seizures, cataracts and MRI imaging performed in your mutant mice?

Q3: Was there any difference in number of spiral ganglion cells?

Q4: Since PEX3 is involved in a ZSD. Are there any domain specific variants that are associated with certain features of the disorder?

Q5: PEX3 has been shown to interact with PEX19. Did you observe any change in its expression in your dataset?

Author Response

“The effect of a Pex3 mutation on hearing and lipid content of the inner ear” is a very well written and comprehensive article.

There are few questions:

Q1: Zellweger spectrum disorder sometimes is associated with mixed hearing loss. Were those aspect of the hearing loss evaluated?

We examined the middle ear of the Pex3tm1a mutant while exposing the cochlea for endocochlear potential recordings but nothing abnormal, such as fluid or debris in the middle ear, was noted. Furthermore, a middle ear defect leads to a wide impact on thresholds affecting all frequencies and this is not what we see – the impact is mostly restricted to higher frequencies in the Pex3tm1a allele. These observations argue against a middle ear defect. We have added a comment about this to the second paragraph of the discussion.

Q2: Were seizures, cataracts and MRI imaging performed in your mutant mice?

We did not see any seizures and we did not carry out MRI imaging on the Pex3tm1a mutants, but in our previous publication (Ingham et al 2019; see Suppl Table 1) we did report results of an extensive phenotypic screen of these homozygous mutants including eye histopathology and DEXA and faxitron imaging. This screen revealed male infertility, reduced alanine and aspartate aminotransferase levels in males, reduced amylase levels in females, increased platelet volume, increased Treg cell and mature B cell percentages, and corneal defects. The eye defects included disruption of the basal layer of the corneal epithelium, loose stromal layers with presence of blood vessels and increased thickness, and fusion of the iris to the lens. These findings were mentioned briefly in the original submitted paper on lines 71-74 where we describe the background information about the mice used, but we have now added additional details about the eye defects here.

Q3: Was there any difference in number of spiral ganglion cells?

We did not count spiral ganglion cells, but as the number of GluR2 puncta below inner hair cells was reduced in homozygous mutants at the 24 and 30 kHz locations at P28 (revised figure 6) this is something that we should follow up in future experiments.

Q4: Since PEX3 is involved in a ZSD. Are there any domain specific variants that are associated with certain features of the disorder?

There are no obvious domains of the PEX3 protein that associate with particular phenotypes, but relatively few have so far been reported.  These include nonsense (R300X) and missense mutations (G331R, D347Y), a 1bp insertion, deletion of an exon leading to a frameshift and loss of a splice acceptor site.  The phenotypes ranged from very severe with death at 19 days old to survival into middle age. Unsurprisingly, the missense mutations tended to be the mildest in impact on phenotype. This summary has been added to the introduction.

Q5: PEX3 has been shown to interact with PEX19. Did you observe any change in its expression in your dataset?

That is an interesting question but we have not looked at it yet. We will plan to do so but cannot do so in the time allowed for revision.

Round 2

Reviewer 1 Report

The authors have responded to the concerns of the first review. I appreciate the attention to detail shown.

To be more clear, it would be of value to the readers (but not required) if there were any data reported on VCLFA and/or plasmalogen levels in the Pex3(tm1a) homozygous mutant mice in  blood or any non-inner related tissue of interest, such as liver. This model may be of interest to a larger scope of readers interested in the role of peroxisome in different organ systems. Thus this information could be of value, especially if it already exists in one form or another.

The concern I raised about Figure 7 (formally Figure 6) involved the SM data. Perhaps, I am misinterpreting the data since it is more common in the peroxisome field to report measurements of the levels of specific endogenous lipids (e.g., C26:0-LPC). Herein in Figure 7, SM22:2 wild type mice > hz Pex3 tma1a mutant mice. However, for SM26:2, SM26:3, and SM26:4, Pex3 tma1a mutant mice were > wild type mice. One would have to assume these are different SM (sphingomyelin) species being measured. The exact species is unknown (i.e., its chemical formula), is that correct? It would seem to be implied since the standards are non-endogenous to the samples.

It would be useful to explain this in simple terms for readers who may not be well-informed about lipidomic technologies.

Overall, the authors have done a fine job revising the text and the above suggestions are meant to improve the impact of the manuscript in the field.

Author Response

We agree that it would be helpful to look at lipids in other tissues in these Pex3 mutant mice, but our current study is focussed on hearing so we have only included the data from lipid analysis of the inner ear. Other tissues could be studied in future experiments.

In the old figure 6, new figure 7, we have only plotted the findings where there was a significant difference between the mutant and the wildtype inner ears, and we included all of the significant lipid species not just those previously associated with peroxisomes. The complete dataset is now provided in Supplementary Table 1, which lists the individual mouse results for all lipids we were able to look for. We mention the results for the lipid C26:0-LPC in the revised discussion. The various lipid species listed (SM22:2, SM26:2, SM26:3, SM26:4) are different species with their chemical composition known, as revealed by the orthogonality of the liquid chromatography system (LC) and the mass accuracy of the mass spectrometer (orbitrap) employed, giving a mass error below 5 ppm. We used the updated LIPID MAPS nomenclature (PMID: 33037133). The SM results may not fit with all of the expectations of a peroxisome disorder but we think it is important to provide the data for others to interpret as well as our own interpretation that we presented in our paper.

Reviewer 2 Report

Congrats on the study.

Author Response

Thank you!